# Metastasis of Uveal Melanoma with Monosomy-3 Is Associated with a Less Glycogenetic Gene Expression Profile and the Dysregulation of Glycogen Storage

**DOI:** 10.3390/cancers12082101

**Published:** 2020-07-29

**Authors:** Siranush Vardanyan, Anton Brosig, Hartmut Merz, Mahdy Ranjbar, Vinodh Kakkassery, Salvatore Grisanti, Aysegül Tura

**Affiliations:** 1Department of Ophthalmology, University of Lübeck, Ratzeburger Allee 160, 23538 Lübeck, Germany; siran1992@mail.ru (S.V.); Anton.Brosig@uksh.de (A.B.); Mahdy.Ranjbar@uksh.de (M.R.); Vinodh.Kakkassery@uksh.de (V.K.); Salvatore.Grisanti@uksh.de (S.G.); 2Reference Center for Lymph Node Pathology and Haematopathology, 23562 Lübeck, Germany; merz@haematopathologie-luebeck.de

**Keywords:** uveal melanoma, monosomy-3, glycogen, glycogenin, insulin resistance, tumor dormancy

## Abstract

The prolonged storage of glucose as glycogen can promote the quiescence of tumor cells, whereas the accumulation of an aberrant form of glycogen without the primer protein glycogenin can induce the metabolic switch towards a glycolytic phenotype. Here, we analyzed the expression of *n* = 67 genes involved in glycogen metabolism on the uveal melanoma (UM) cohort of the Cancer Genome Atlas (TCGA) study and validated the differentially expressed genes in an independent cohort. We also evaluated the glycogen levels with regard to the prognostic factors via a differential periodic acid-Schiff (PAS) staining. UMs with monosomy-3 exhibited a less glycogenetic and more insulin-resistant gene expression profile, together with the reduction of glycogen levels, which were associated with the metastases. Expression of glycogenin-1 (Locus: 3q24) was lower in the monosomy-3 tumors, whereas the complementary isoform glycogenin-2 (Locus: Xp22.33) was upregulated in females. Remarkably, glycogen was more abundant in the monosomy-3 tumors of male versus female patients. We therefore provide the first evidence to the dysregulation of glycogen metabolism as a novel factor that may be aggravating the course of UM particularly in males.

## 1. Introduction

The prevention and therapy of uveal melanoma (UM) metastases remain as an unmet clinical need, which sadly leaves the patients an average survival time of less than six months following the diagnosis of such systemic lesions [1,2]. The metastases develop in approximately half of the UM patients mainly in the liver usually 5–10 years after the successful local control of the primary tumor, with a latency of up to 40 years [3,4,5]. This pattern suggests that the tumor cells have already started to disseminate at the time of initial diagnosis and could survive in the extraocular environment for many years possibly in a dormant state. The most critical prognostic factor for UM is the loss of one copy of chromosome 3 (monosomy-3), with the occurrence of metastatic disease almost exclusively in the patients having this anomaly in their primary tumor [4,6,7]. Likewise, the presence of monosomy-3 in the circulating melanoma cells (CMCs) and metastasized UMs was associated with a higher metastatic risk and disease progression rate, respectively [8,9]. Monosomy-3 therefore serves as an independent prognostic factor for shorter survival following the diagnosis of metastases and the UMs with this anomaly are classified as high-risk tumors [4,7]. UM exhibits a slight predominance among males, with 4.9 cases per million compared to females at 3.7 cases per million [7,10,11]. Men also tend to have earlier and more frequent metastases with a worse survival rate [12,13]. However, the molecular mechanisms underlying the unfavorable prognosis in males or the growth advantage conferred by monosomy-3 that enables the faster recovery of the disseminated UM cells from dormancy are not sufficiently elucidated for the development of efficient and preventive anti-metastatic therapies.

Remarkably, the presence of monosomy-3 in primary UMs was positively correlated with the metabolic activity of these tumors in positron emission tomography (PET) scans [14], which indicates a higher rate of glucose influx into the UM cells with monosomy-3. However, it is not known how much of the intracellular glucose is metabolized immediately for cellular growth versus preserved for later usage in these tumors. In animal cells, glucose is mainly stored in the form of glycogen as an energy reserve. This process is not only restricted to the normal cells in the liver, muscle, intestine, or brain, but also observed in many types of tumor cells regardless of their origin [15,16,17]. The glycogen reserves indeed enable the survival of cancer cells under hypoxia, such as in the rapidly growing solid tumors with an inadequate blood supply [15,18]. However, this rescue apparently occurs at the expense of cell growth, since the amount of normal glycogen in tumor cells was found to be inversely correlated with the proliferation rate [17,19]. An increase in glycogen synthesis may therefore be informing the tumor cells that the environmental conditions are not suitable for excessive growth and energy consumption. This protective attempt to ensure the survival under more drastic, nutrient-limiting conditions consequently induces the quiescence and premature senescence of cancer cells. In contrast, the tumors with “unstable” glycogen can continue to utilize glucose for the production of ATP, nucleotides, and amino acids, which would sustain their faster growth [15,19].

The synthesis of glycogen is initiated by the core protein glycogenin-1, which dimerizes and autoglucosylates itself to form a short glucose polymer [15,16]. In the absence of this primer protein, its larger isoform glycogenin-2 can be upregulated in some cells as a compensation [20,21]. The short glucose polymer is extended further by the addition of glucose molecules via the glycogen synthase enzyme. When this primary oligosaccharide chain reaches a length of 12 glucose units, it continues to expand by branching via the glycogen branching enzyme. The elongation of glycogen therefore proceeds in balance with its branching, which is essential for improving the solubility and storage of this macromolecule in the cytoplasm [15,16].

Interestingly, we have noticed that the genes encoding glycogenin-1 (GYG1, NCBI Gene ID: 2992) and the glycogen branching enzyme (GBE1, Gene ID: 2632) are localized to chromosome 3q24 and 3p12.2, respectively, while the gene for the complementary isoform glycogenin-2 (GYG2, Gene ID: 8908) resides on chromosome Xp22.33. Moreover, the BAP1 tumor suppressor, which is encoded by a gene on chromosome 3p21.1 and the loss of which is strongly associated with UM metastases [4,7], may also be positively regulating glycogen synthesis as demonstrated by the depletion of glycogen in the livers of mice with a conditional *Bap1* deletion [22]. In contrast, the glycogen synthase kinase 3B enzyme (GSK3B, Locus: 3q13.33) acts as a negative regulator of glycogen synthesis by phosphorylating and thereby inactivating the glycogen synthase [15,23]. The silencing of the glycogenin-1 gene in *Gyg1*-knockout mice has also resulted in the unexpected accumulation of an aberrant type of protein-free glycogen with a larger molecular size, which led to the metabolic switch of muscle cells toward a more glycolytic and less oxidative phenotype despite the maintenance of normal mitochondria [24]. However, it is not explored yet, whether the depletion of the glycogenin isoforms GYG1 or GYG2 induces such a glycolytic switch in tumor cells. The preference of glycolysis over the oxidative phosphorylation in mitochondria, which is commonly known as the Warburg effect, is indeed a remarkable feature of various cancer cells to generate energy and other biosynthetic materials from glucose, which may be accounting for an aggressive growth rate [25,26,27]. Yet, it has remained unknown whether the UM cells can store the glucose as glycogen or the presence of monosomy-3 is associated with a gender-specific difference in this event that may be influencing the clinical outcome.

The detection of glycogen in tissues can be performed by a differential periodic acid-Schiff (PAS) staining with and without the glycogen-degrading enzyme amylase [15,21]. The normal PAS stainings of UMs have indeed revealed striking differences between benign and malignant tumors, with the presence of extracellular loops and networks in the latter group. However, the interpretation of these staining patterns was mainly focused on the extracellular matrix in vasculogenic mimicry events, without any reference to glycogen [4,7,28]. Since a differential PAS staining has not been reported on UM so far, it is not known, how much of these staining patterns reflect the amount of glycogen in these tumors.

To gain more insight into these aspects, we initially analyzed the mRNA levels of *n* = 67 genes involved in the synthesis or breakdown of glycogen (Appendix A) in the UM cohort of the TCGA study with respect to the histological and clinical prognostic factors. We also validated the differentially expressed genes in an independent cohort from the Gene Expression Omnibus (GEO) database with *n* = 57 UM patients. In addition, we performed a differential PAS staining to estimate the glycogen content in the primary UMs of *n* = 30 patients operated in our clinic with regard to the monosomy-3 status in the primary tumor and CMCs, patient gender, and clinical outcome. Our findings highlight the dysregulation of glycogen storage as a novel pathomechanism that may be worsening the prognosis of UM with monosomy-3 particularly in the male patients.

## 2. Results

### 2.1. Expression of the Genes Involved in Glycogen Synthesis or Breakdown with Regard to the Monosomy-3 Status in the TCGA Cohort

The clinical data of the patients in the UM cohort of the TCGA study have been thoroughly described elsewhere [29]. Briefly, the cohort included *n* = 80 patients with a median age of 61.5 years and of whom 56.3% (*n* = 45) were males. The patients have undergone enucleation or tumor resection without prior irradiation. The gene expression in the primary tumors with regard to the monosomy-3 status was evaluated using the data of *n* = 77 of the 80 patients with complete information on the copy numbers of chromosome 3p and 3q in the cBioPortal database, whereas the remaining three patients with missing data were excluded from this analysis.

We initially identified *n* = 67 genes encoding proteins involved in glycogen synthesis or breakdown based on a literature and databank search using the platforms PubMed, Gene, OMIM, and UniProt (Appendix A) [15,16,22,23,30,31,32,33,34,35,36,37,38,39,40,41,42,43,44,45,46,47,48,49,50,51,52,53]. We then calculated the median mRNA expression of these genes with regard to the monosomy-3 status with or without correction for the gene copy number. Using this approach, we could determine that *n* = 22 of the 67 genes (32.8%) exhibited a differential expression pattern in the monosomy-3 tumors regardless of the gene copy number (Table 1, Appendix A). The unbiased gene set enrichment analysis demonstrated that these 22 genes were mainly associated with the Gene Ontology terms “energy reserve/glycogen/glucan metabolic processes.” Likewise, the major biological pathways involving these genes included the carbohydrate and glucose metabolism, as well as the glycogen synthesis (Figure 1).

Among the 22 genes, the transcripts of *n* = 13 genes were significantly lower in the monosomy-3 tumors (ATG7, BAP1, EPM2A, GBE1, GSK3B, GYG1, GYG2, INSR, IRS2, IRS4, PHKA2, PPP1R3B, PPP1R3C; Table 1, Appendix A), whereas the remaining nine genes (CALM1, CALM2, G6PC3, GFPT1, PCK1, PGM2, PGM5, PPP1R14C, PYGM) exhibited a higher expression in the tumors with monosomy-3 (Table 1, Appendix A). Of the 13 genes with a lower expression in the monosomy-3 tumors, *n* = 3 genes were directly involved in de novo glycogen synthesis (GBE1 on chromosome 3p12.2, GYG1 on chromosome 3q24, GYG2 on chromosome Xp22.33) [15,16], and *n* = 8 genes were reported to be exerting a glycogenetic effect [15,16,22,31,32,33], whereas the remaining two genes (GSK3B on chromosome 3q13.33 and PHKA2 on chromosome Xp22.13) were acting as negative regulators of glycogen synthesis (Table 1, Figure 2) [15,16,23]. The upregulated genes (*n* = 9) in the monosomy-3 tumors include seven genes with a negative effect on glycogen synthesis [15,16,34,35,36], whereas the remaining two genes (PGM2 and PGM5) were involved in both the production and breakdown of glycogen (Table 1, Figure 3) [16,38]. The median fold change of the downregulated genes was found to be −1.62 (range: −1.15–−9.85). Likewise, the median fold change of the upregulated genes was 1.74 (range: 1.41–25.99, Table 1). The downregulated genes were mainly related to the glycogen/glucan/glucose metabolism, glycogen synthesis, and glycogen storage diseases. Likewise, the upregulated genes were associated with the carbohydrate and energy reserve metabolic processes, glucose breakdown, and glycogen lysis (Figure 1).

### 2.2. Expression of the Genes Involved in Glycogen Metabolim with Regard to the Prognostic Factors and Clinical Outcome in the TCGA Cohort

The mRNA levels of the 22 genes that were differentially expressed in the monosomy-3 tumors were analyzed with regard to the histopathological and clinical prognostic factors as well as the disease-specific survival in all the patients of the TCGA-UM cohort (*n* = 80) with available data (Appendix A).

The lower expression of the 13 genes that were downregulated in the monosomy-3 tumors (Table 1) was significantly associated with reduced BAP1 levels. Except for GSK3B, the transcription of the remaining 12 genes was also decreased in the primary tumors of the patients who have developed metastases. Progression and disease-related death were correlated with the lower expression of *n* = 11 and 12 genes, respectively, excluding IRS4 and GSK3B in the former and GYG1 in the latter group. Older patients tended to have less IRS4 levels (*p* = 0.03). Expression of GYG2 (Locus: Xp22.33) was significantly lower in the male patients (*p* < 0.01). Tumors with the largest basal diameter above 13 mm exhibited a decline in BAP1, EPM2A, GYG1, GYG2, and PPP1R3C levels, whereas only EPM2A was negatively associated with an increased tumor thickness. Reduced expression of EPM2A, PPP1R3B, and PPP13C was also correlated with an advanced clinical stage. Extrascleral extension was observed in the tumors with lower levels of EPM2A, INSR, IRS4, PPP1R3B, PPP1R3C, and PHKA2. Tumors with an epithelioid versus spindle morphology exhibited a reduction in the mRNA of ATG7, BAP1, EPM2A, GYG2, INSR, IRS2, PPP1R3B, PPP1R3C, and PHKA2. Closed connective tissue loops were present in the tumors with a lower expression of ATG7, BAP1, GYG1, INSR, IRS4, PPP1R3B, and PPP1R3C. Mitotic count was elevated in the tumors with a decrease in GYG2 and PPP1R3C levels. A heavy infiltration with lymphocytes was observed in the tumors with a reduction in GBE1, IRS2, PPP1R3B, PPP1R3C, and PHKA2. Likewise, a heavier infiltration with macrophages and degree of pigmentation were detected in the tumors with a lower expression of EPM2A, GBE1, IRS4, PPP1R3B, PPP1R3C, and GSK3B. Tumor necrosis was negatively correlated with the levels of PPP1R3C. Patients with a higher body-mass index (equal to or above 25.0) exhibited a reduction in the IRS2 transcription. No significant association was detected between the eye color and the analyzed genes that were downregulated in the monosomy-3 tumors (Appendix A).

The higher expression of the nine genes that were upregulated in the monosomy-3 tumors (Table 1) was significantly correlated with reduced BAP1 levels. Except for PGM5, the remaining eight genes were also elevated in the patients who have succumbed to metastatic disease. The genes CALM1, CALM2, GFPT1, PCK1, PGM2, PPP1R14C, and PYGM were further associated with the presence of closed connective tissue loops, disease progression and metastases. PCK1 and PPP1R14C were elevated in the tumors with the largest basal diameter above 13 mm and an advanced clinical stage. Higher PCK1 levels were also correlated with an elevated mitotic count (*p* = 0.01). Extrascleral extension was observed in the tumors with an increased expression of GFPT1 and PPP1R14C. Tumors with a more epithelioid morphology exhibited an upregulation of CALM1, CALM2, GFPT1, PCK1, PPP1R14C, and possibly PGM5. A moderate to heavy infiltration with lymphocytes was observed in the tumors with an increase in CALM1, PYGM, and PGM5, whereas the heavier infiltration with macrophages and degree of pigmentation were positively correlated only with the expression of G6PC3. No significant association was detected between the age, gender, eye color, tumor thickness, tumor necrosis, body-mass index, and any of the genes that were elevated in the monosomy-3 tumors (Appendix A).

### 2.3. Validation of Gene Expression in an Independent UM Cohort

The differentially expressed genes in the TCGA study were validated in an independent UM cohort from Cleveland, OH available at the GEO Database (accession number: GSE44295). The validation cohort consisted of *n* = 57 patients, with *n* = 32 males (56.1%). Metastases have developed in *n* = 24 patients (42.1%). Since the data on the monosomy-3 status of these tumors were not accessible, the patients were classified according to the development of metastases and gender to compare the median gene expression in these groups. Multiple probes were present for the EPM2A, GYG2, and PGM5 genes (Appendix A).

Of the 22 genes analyzed, *n* = 5 genes were expressed at significantly lower levels in the metastatic patients of the validation cohort and included ATG7, BAP1, EPM2A, GYG2, and PPP1R3C, which all function as the positive regulators of glycogen synthesis. GYG2 was the only gene that exhibited a gender-specific expression pattern, with the downregulation in the male patients. After Bonferroni correction, GYG2 remained as the only gene with a significantly lower expression in the metastatic or male patients (Figure 4, Appendix A). Except for ATG7, the reduced expression of the remaining validated genes was associated with a lower overall survival rate in the TCGA cohort (Figure 5).

### 2.4. Analysis of Glycogen Storage in the Primary UM of Our Patients with Regard to the Monosomy-3 Status and Clinical Factors

To determine the influence of monosomy-3 on the glycogen storage capacity of the UM cells, we also performed a differential PAS staining with or without the glycogen-degrading enzyme amylase and estimated the glycogen content in the primary tumor samples from *n* = 30 UM patients operated in our clinic. Our patient cohort exhibited a median age of 69 years and consisted of *n* = 16 (53.3%) males. Irradiation was performed in *n* = 16 patients (53.3%) at a median of two months (range: 1–16 months) prior to the enucleation or tumor resection. Monosomy-3 was detected in 46.7% (*n* = 14 of 30) of the primary tumors. CMCs were isolated from *n* = 29 of the 30 patients. Except for two patients with metastases in the liver or bone at the initial presentation, all the remaining patients were diagnosed with primary UM without metastases. The clinical data of the patients and the histological features of the tumors are summarized in Table 2.

Images of the entire tumor area were acquired for all the samples with and without amylase pretreatment. An objective quantification of the PAS staining intensity was then performed on all the images by determining the pixel intensity of the magenta layer. To estimate the glycogen levels, the difference in the staining intensity of the untreated and the corresponding, amylase-pretreated samples of each tumor was calculated and expressed as the percentage of the intensity in the untreated tumor.

The objective quantifications demonstrated a significantly reduced level of glycogen in the tumors with a higher extent of monosomy-3 and lower nuclear BAP1 expression (Figure 6, Table 2). Although the pretreatment with amylase also led to the weakening of the PAS-staining of extracellular arcs, we did not observe the complete disappearance of these structures (Figure 6a,b, not quantified).

Metastases have developed in eight more patients during the follow-up time of 2–9 years, resulting in the presence of systemic lesions in *n* = 10 of the 30 patients (33.3%, *n* = 5 females and males each). The primary tumors of all the metastatic patients were classified as having monosomy-3. The percentage of CMCs with monosomy-3 was also elevated in the metastatic versus non-metastatic patients (Median: 37.5 vs. 10.8%, respectively, *p* = 0.03, Mann–Whitney U test). The estimated glycogen amount was inversely correlated with the presence of metastases (*p* = 0.02). Patients with lower glycogen levels also tended to be diagnosed with UM at a younger age, but this effect failed to reach significance (*p* = 0.07). No association was observed between the glycogen levels and the remaining parameters including the affected eye, irradiation, tumor size, cytoplasmic BAP1 expression, presence or number of CMCs, presence of monosomy-3 in the CMCs, and patient gender (Table 2).

Remarkably, the further stratification of the female and male patients with regard to the monosomy-3 status, irradiation, or metastases revealed several gender-specific differences. For instance, the levels of glycogen were higher in the monosomy-3 tumors of male versus female patients (*n* = 7 patients each, *p* = 0.03, Figure 7, Appendix A). Glycogen was also more abundant in the non-irradiated tumors of male patients compared to females, but this effect failed to reach significance (*n* = 7 patients per group, *p* = 0.05). In contrast, the percentage of CMCs with monosomy-3 was lower in the male patients (*n* = 14 females, *n* = 13 males, *p* = 0.02). CMCs with monosomy-3 were detected at a similar extent in the female patients without or with metastases (Median: 33.3% for each group, *n* = 9 and 5 patients, respectively, *p* = 0.74). However, the male patients with metastases exhibited significantly more CMCs with monosomy-3 compared to the non-metastatic patients (Median: 43.8% vs. 0.0%, *n* = 9 and 4 patients, respectively, *p* = 0.01). Accordingly, the prevalence of CMCs with monosomy-3 was higher in the female versus male patients without metastases (median: 33.3% vs. 0.0%, *n* = 9 patients per group, *p* = 0.01, Figure 7). No remarkable difference was observed in the number of CMCs with regard to the patient gender. The median time until the development of metastases tended to be eight months shorter in the male versus female patients, but this effect remained below the level of significance (Median: 17.5 versus 25.5 months, respectively, *p* = 0.69, Mann–Whitney U test, *n* = 4 patients per group).

## 3. Discussion

Metabolic reprogramming of solid tumors enables their adaptation to the environmental changes and the development of resistance to adverse factors such as hypoxia and energy deprivation. An early response of the cancer cells to hypoxia is the storage of glucose in the form of glycogen, followed by the gradual breakdown of this energy reserve, which enables tumor survival [15,54]. In contrast, the prolonged accumulation of glycogen and the impairment of glucose mobilization can induce the premature senescence of tumor cells and inhibit growth [17,19]. Remarkably, the chromosome 3 harbors several genes that are directly involved in glycogen metabolism, but it has remained unknown whether the UM cells can store glycogen and how the presence of monosomy-3 influences this process. In this study, we report the differential expression of *n* = 22 genes that regulate glycogen dynamics and provide the first evidence to the dysregulation of glycogen storage in the primary UMs with monosomy-3, which was associated with a worse prognosis.

Among the 13 genes that exhibited a lower expression in the monosomy-3 tumors, *n* = 11 genes were acting as positive regulators of glycogen synthesis and include ATG7, BAP1, EPM2A, GBE1, GYG1, GYG2, INSR, IRS2, IRS4, PPP1R3B, and PPP1R3C. The ATG7 gene, which resides on chromosome 3p25.3, encodes the autophagy-related 7 protein, and its selective inhibition could prevent the accumulation of glycogen in autophagy-deficient mice [15]. Likewise, the targeted deletion of the BAP1 tumor suppressor (Locus: 3p21.1) resulted in the depletion of glycogen from the liver of knockout mice [22]. In our study, we have also observed lower glycogen levels in the primary UMs with a reduced expression of nuclear BAP1, providing additional support to the recently described role of BAP1 in cellular metabolism. GYG1 and GBE1 are further genes on chromosome 3, which encode the glycogenin-1 and glycogen branching enzymes that are directly engaged in de novo glycogen synthesis, while the GYG2 gene on chromosome Xp22.33 codes for the complementary isoform glycogenin-2 [15,16]. The Laforin protein encoded by the EPM2A gene on chromosome 6q24.3 prevents the excessive phosphorylation and thereby the aberrant branching of glycogen [16]. In our study, the lower expression of EPM2A was also associated with several factors that adversely affect the prognosis in the TCGA cohort, such as a larger tumor size, advanced clinical stage, extrascleral extension, and a heavier infiltration with macrophages. It might be therefore very informative to focus further on the contribution of the EPM2A-depletion to the unfavorable prognosis of the UM tumors with the 6q loss [7] and dysregulation of glycogenesis.

Interestingly, the remaining glycogenetic genes which were downregulated in the monosomy-3 tumors are all involved in the transmission of insulin-dependent signalling. For instance, the INSR gene encodes the insulin receptor whereas the insulin receptor subtrates 2 and −4 (IRS2 and IRS4) belong to the immediate downstream effectors of INSR [31,32,33]. Likewise, the PPP1R3B and PPP1R3C proteins serve as the glycogen-binding regulatory subunits of the protein phosphatase-1 (PP1) complex, which enhances glycogen synthesis in response to insulin [15,16]. The downregulation of these genes in the UMs with monosomy-3 therefore indicates the impairment of normal insulin signaling in such samples. The lower expression of the IRS2 gene in the patients with a higher body-mass index also suggested the presence of a systemic metabolic disorder in a subgroup of the UM patients. Remarkably, the chromosome 3 was already reported to be exhibiting the strongest linkage to the serum insulin concentrations and fasting insulin resistance index [55]. The systemic resistance to insulin, as determined by the decreased circulation of the insulin-sensitizing hormone adiponectin, was indeed correlated with a worse prognosis in the patients with choroidal nevi or UM [56]. We have also recently reported the lower expression of adiponectin (Locus: 3q27.3) and its receptor Adipor1 in the UM samples with monosomy-3, suggesting the occurrence of a local insulin resistance in such tumors [57]. In this study, we provide further support to the existence of an insulin-resistant state in the UMs with monosomy-3, which was associated with the onset of metastasis and reduced survival rate. Additional studies on the mediators of the insulin-dependent signalling pathways at the protein level and their functional consequences as well as the presence of metabolic disorders would therefore be invaluable for the identification of novel therapeutic targets that may be modified at an earlier stage in the patients with UM or melanocytic lesions.

The remaining two genes with a lower expression in the monosomy-3 tumors of the TCGA cohort (GSK3B and PHKA2) function as the negative regulators of glycogen synthesis. GSK3B on chromosome 3q13.33 encodes the glycogen synthase kinase beta enzyme, which phosphorylates and thereby inactivates the glycogen synthase [15,23]. However, the GSK3B protein can interact with other downstream targets and regulates diverse cellular events, such as the activation of the tumor suppressor p53 [58,59]. The loss of such additional functions might therefore have contributed to the worse prognosis associated with a lower expression of GSK3B in the TCGA cohort. Likewise, the PHKA2 gene on chromosome Xp22.13 encodes the alpha subunit of the phosphorylase kinase enzyme that promotes glycogen lysis [16]. In contrast to the TCGA cohort, the median expression of the PHKA2 and GSK3B genes was higher in the metastatic or male patients of our validation cohort, but this effect failed to reach significance (Appendix A). Further analysis of the PHKA and GSK3B expression at the protein level would therefore be more informative in defining the role of these genes in the glycogen metabolism of UM cells.

We have also detected nine genes with a significantly higher expression in the monosomy-3 tumors of the TCGA cohort. Among these, *n* = 7 genes are involved in the negative regulation of glycogen synthesis and include CALM1, CALM2, G6PC3, GFPT1, PCK1, PPP1R14C, and PYGM, whereas the remaining two genes (PGM2 and PGM5) execute a dual role in both the production and degradation of glycogen via the interconversion of glucose-6-phosphate and glucose-1-phosphate [16,38]. The breakdown of glycogen can occur either in the cytosol or lysosomes. The former process is catalyzed by the coordinated action of the glycogen phosphorylase and glycogen debranching enzymes while the latter event is initiated by the lysosomal acid alpha-glucosidase (GAA) [15,16]. In our study, the genes encoding the muscle isoform of glycogen phosphorylase (PYGM) and GAA were both upregulated in the monosomy-3 tumors, but this effect remained significant after Bonferroni correction only for the PYGM gene (Appendix A). Activation of glycogen phosphorylase is mediated by the phosphorylation via the glycogen phosphorylase kinase enzyme that involves the CALM1 and CALM2 subunits [16]. PPP1R14C can also promote the glycogen phosphorylase activity by inhibiting the PP1 complex that dephosphorylates PYGM [15,16,36]. G6PC3 encodes a catalytic subunit of the glucose-6-phosphatase enzyme, which executes the final step of the glycogen breakdown to generate free glucose [16]. The overexpression of the PCK1 gene in liver could induce an insulin-resistant phenotype with a reduction in glycogen content [35]. Likewise, the GFPT1 gene is associated with familial hyperinsulinemia and acts as a negative regulator of glycogen synthesis [34]. The differential expression of these genes in the TCGA cohort therefore provides further support to a less glycogenetic and more insulin-resistant phenotype in the monosomy-3 tumors.

Our validation analysis on a smaller cohort with *n* = 57 patients confirmed the downregulation of the glycogenetic genes ATG7, BAP1, EPM2A, GYG2, and PPP1R3C in the primary tumors of the UM patients who later developed metastases. GYG2 (Locus: Xp22.33) was also the only gene that exhibited a gender-specific difference in both cohorts, with a higher expression in females, suggesting the escape of this gene from X-chromosome inactivation. In females, the transcription of genes on one of the X-chromosomes is usually silenced to compensate for the dosage imbalance of the X-chromosome compared to males. However, up to one third of the X-chromosomal genes that are mainly localized on the distal short arm (Xp) escape this inactivation, resulting in a stronger expression in females [60,61]. The GYG2 gene indeed resides more distally at the Xp22.33 domain and was capable of escaping X-chromosome inactivation as opposed to PHKA2 (Locus: Xp22.13) [62]. The upregulation of GYG2 expression in the TCGA cohort was also inversely associated with several other prognostic factors including the tumor basal diameter, epithelioid morphology, closed connective tissue loops, and mitotic count. The GYG2 gene has a shorter homolog in the Y-chromosome, which is 3′ truncated and unable to generate a functional protein (GYG2P1, Locus: Yq11.221, Gene ID: 352887) [63]. The less efficient upregulation of the GYG2 isoform in the male versus female patients in response to the loss of GYG1 (Locus: 3q24) in the monosomy-3 tumors may therefore be contributing to the worse prognosis of the former group by impairing the synthesis of normal glycogen, which deserves further investigation.

In summary, our results provided the first evidence for a less glycogenetic and more insulin-resistant gene expression profile in the monosomy-3 tumors that was associated with the development of metastases and a reduced survival rate. Since the fold changes of the up- and downregulated genes exhibited the median values of 1.74 and −1.62, respectively, these relatively minor alterations might have remained “under the radar” in the previous, large-scale studies that have used high cut-off values [29]. Nevertheless, the co-occurrence of such small changes that affect multiple steps of the glycogen metabolism might be exerting a cumulative effect on this event. Such an additive outcome was indeed reported in a previous gene set enrichment study, which demonstrated the downregulation of genes involved in oxidative phosphorylation in the muscle biopsies of human diabetic patients, with an average reduction of 20% per gene [64]. Since the mRNA levels may not always be reflecting the amount of active proteins [65], further studies at the protein level would also be invaluable for determining the functional consequences of this altered gene expression profile on the glycogen metabolism with regard to the monosomy-3 status.

Our differential PAS-stainings provided additional support to the findings of the gene expression analysis and demonstrated lower levels of the amylase-sensitive glycogen in the primary UMs with monosomy-3, which was also associated with the development of metastases. However, our sample size was not sufficient to draw a reliable conclusion on several aspects, such as the influence of irradiation on the glycogen metabolism of UM cells. We were also not able to determine whether the formation of aberrant glycogen fibers and their possible exocytosis [66] contributed to the extracellular vasculogenic mimicry patterns, which are usually visualized by a normal PAS staining without amylase pretreatment. Abnormal glycogen fibers with a less branched structure are indeed the hallmarks of several glycogen storage disorders and induce pathogenesis by accumulating as polyglucosan bodies that are more resistant to amylase digestion [67,68]. Remarkably, the extracellular polyglucosan bodies can exhibit a tendency to congregate in the vicinity of blood vessels [69,70] possibly as an attempt to facilitate their disposal from the affected tissues. In our study, we have observed a weakening in the PAS-staining of the extracellular arcs after amylase pretreatment, but we were not able to reliably quantify these changes. We could therefore not ascertain whether the extracellular glycogen with a varying sensitivity to amylase digestion is involved in vasculogenic mimicry events. Although our gene expression analysis suggested the association of a less glycogenetic phenotype with the presence of closed loops and networks, we could not evaluate the contribution of the encoded proteins to the vasculogenic mimicry, either. Future studies with the purification of glycogen from native tumors may therefore enable the structural characterization of this macromolecule and the associated proteins, which would provide valuable insight into the possible involvement of glycogen metabolism in vasculogenic mimicry particularly in the monosomy-3 tumors.

Interestingly, the further stratification of our patients according to the monosomy-3 status or metastases revealed intriguing differences that were gender-dependent. For instance, the glycogen levels were significantly higher in the monosomy-3 tumors of male versus female patients. Although this finding may initially appear very contradictory to our results on the gender-specific expression profile of GYG2, we would like to underline a limitation of our study. Using the differential PAS staining, we could only estimate the amount of amylase-sensitive glycogen, and we were not able to further differentiate between the structural features of this macromolecule. Aberrant forms of glycogen can indeed be generated due to the impairment of certain genes such as the GYG1 on chromosome 3q24, which encodes the glycogenin-1 protein that serves as the primer of glycogen synthesis by catalyzing the addition of glucose molecules on to itself [15,16]. Rather than preventing glycogen production, the silencing of *Gyg1* has surprisingly led to the over-accumulation of an aberrant type of glycogen with a larger size. This abnormal glycogen was also devoid of a protein primer, suggesting that a complementary protein substrate such as GYG2 was not utilized for its synthesis in the *Gyg1*-knockout mice. As opposed to the humans, mice lack the gene for the GYG2 isoform, which might have contributed to the formation of a protein-free glycogen in the *Gyg1*-depleted animals [24]. These findings therefore strongly support our observations and provide an explanation for the peculiar, gender-specific differences in the glycogen levels of our UM patients. As expected, the loss of one copy of the *Gyg1* gene in the monosomy-3 tumors would impair the glycogen synthesis and account for the significantly lower levels of this macromolecule. The female patients with monosomy-3 may be compensating for the deficiency of GYG1 to a certain extent by the upregulation of the *Gyg2* homolog on chromosome X, while the male patients with monosomy-3 may be more restricted in this aspect, as suggested by the gene expression profiles. The monosomy-3 tumors of the male patients would therefore be more inclined to store the glucose as a protein-free glycogen with an abnormal structure and accumulation rate compared to the females, that needs to be characterized in more detail.

Remarkably, the aberrant, protein-free glycogen in the *Gyg1*-knockout mice could also induce functional consequences, such as the metabolic switch of the muscle cells towards a more glycolytic rather than oxidative phenotype despite the maintenance of normal mitochondria [24]. Such a glycolytic switch is also the major feature of various tumor cells, which enables the utilization of glucose for the production of not only energy, but also other biosynthetic materials required for cell growth. However, to benefit from the glycolytic switch efficiently, the cancer cells need to sustain a massive amount of glucose uptake [25,26,27]. Notably, the UM cells with monosomy-3 may be more capable of a higher rate of glucose influx, as suggested by their elevated PET-scan activity [14]. Yet, it is still not known, how the intracellular glucose is metabolized in the UM cells and whether monosomy-3 induces a gender-specific effect on this process. In our study, the prevalence of CMCs with monosomy-3 was higher in the female versus male patients who did not develop metastases, suggesting that the CMCs of the female patients may be possessing less metastatic potential despite the presence of monosomy-3. In contrast, the male patients with monosomy-3 positive CMCs were significantly more likely to develop metastases. The median time to the development of metastases also tended to be shorter in the male patients, although this effect did not reach significance at our low sample size. A possible reason accounting for the more aggressive growth potential of the male UM cells with monosomy-3 may indeed be the storage of excessive glucose as an aberrant, protein-free glycogen due to the deficiency of both GYG1 and GYG2 on chromosomes 3 and X, respectively. The over-accumulation of this abnormal glycogen might then be inducing a glycolytic switch that would enable the faster growth of the male UM cells with monosomy-3 particularly when the extracellular glucose is highly abundant such as during hyperglycemia, which urgently deserves further investigation.

Sustained hyperglycemia is indeed one of the factors that impairs the insulin-mediated regulation of glucose homeostasis and glycogen synthesis in the muscle and liver cells [71,72]. Remarkably, the liver plays a crucial role in the maintenance of glucose homeostasis by serving as the major glucose storage site in the body. The liver responds to the insulin- and adiponectin-mediated signaling by terminating the release of glucose from its glycogen reserves [73,74,75,76]. The gradual worsening of the systemic insulin resistance and adiponectin depletion would therefore create a more hyperglycemic environment due to the excessive glucose release from the liver. Under such hyperglycemic conditions, the disseminated UM cells with monosomy-3 or their dormant micrometastases in the liver may become more activated due to their impaired ability to store glucose as normal glycogen. The monosomy-3 positive UM cells of the male patients may acquire a further growth advantage by the induction of a glycolytic switch via the accumulation of an aberrant form of protein-free glycogen as suggested above. The elderly male patients may already be more vulnerable to the development and consequences of insulin resistance owing to the lower circulating levels of adiponectin [77] and the higher bioavailability of the insulin-like growth factor-1 [78], which is mainly produced by the liver as an insulin-mimetic hormone, particularly under insulin resistance [79,80], and considered to be the major chemoattractant for the liver metastases of UM [81,82]. The dormant UM cells with monosomy-3 may thereby be benefiting from an abnormal glucose release from the liver in a gender-dependent manner and growing unproportionally faster via a glycolytic switch, which may also be accounting for the very short survival time of the UM-patients with clinically detectable (macro)-metastases in the liver. Taking immediate actions against hyperglycemia and insulin resistance might therefore be a valuable and readily available strategy to retard the development of lethal metastases in the UM patients, which, we believe, deserves urgent attention and widespread implementation in clinical practice.

## 4. Materials and Methods

### 4.1. Patient Selection

A total of *n* = 30 consecutive patients who were diagnosed with UM between December 2009 and January 2018 at the Department of Ophthalmology, University of Lübeck, Germany, were included in the study. The diagnosis of UM was performed by a specialized ophthalmologist via clinical and ultrasound examination. The study was approved by the local ethic committee of the University of Lübeck (File number: 10–200, Date of approval: 17 December 2010) and conforms to the guidelines of the Declaration of Helsinki of 1975, revised in 2013. All patients received an explanation about the nature and possible consequences of the study and gave informed consent before their inclusion.

Standardized A and B scans (I3 eyecubed System-ABD, Ellex Inc., Sacramento, CA, USA) and ultrasound biomicroscopy (VuMax II, Sonomed Inc., New Hyde Park, NY, USA) were performed to determine the size of the tumor, exact intraocular localization, and ciliary body involvement. The metastatic status was evaluated by liver function tests (alkaline phosphatase, aspartate aminotransferase, alanine aminotransferase, bilirubin), ultrasound of the abdomen, and the computer tomography of the chest and abdomen.

### 4.2. Analysis of Gene Expression in the UM Cohorts of the TCGA Study and GEO Database

Gene expression data of the UM cohort of the TCGA study were downloaded from the Xena platform of the University of California Santa Cruz (Santa Cruz, CA, USA, https://xena.ucsc.edu/) using the Genomic Data Commons (GDC) TCGA Ocular Melanoma dataset, which was available in log_2_(FPKM-uq+1) units (FPKM-uq: Fragments aligning per kilobase of transcript per million mapped reads normalized to upper quartile). The graphical summaries of the genomic alterations, clinical factors, and gene expression heatmaps were generated using the “TCGA PanCancer Atlas” dataset in the cBioPortal resource (https://www.cbioportal.org/) [83,84]. An unbiased analysis for gene set enrichment was performed by using the Genevestigator software (version 7.6.0) [85] and selecting the full lists of human Gene Ontology annotations (*n* = 21,466 gene sets with *n* = 18,455 genes; http://geneontology.org/) [86,87] or human Reactome annotations (*n* = 2018 gene sets with *n* = 9717 genes; https://reactome.org/) [88] as the background collection of gene sets. Kaplan-Meier curves for overall survival were generated using the Xena platform.

Data validation was performed by analyzing the mRNA expression of the genes of interest in an independent cohort from the GEO database (https://www.ncbi.nlm.nih.gov/geo). The validation cohort consisted of *n* = 57 patients with available data on gender and metastatic status (Accession number: GSE44295). RNA analysis had been performed on snap-frozen primary UM tissues from enucleated eyes using the The HumanHT-12 v3 gene expression microarray (Illumina, San Diego, CA, USA).

### 4.3. Differential PAS Staining

Samples of the primary tumor were fixed in 4.5% formalin and embedded in paraffin. Sections with a thickness of 6 µm were collected onto Superfrost Ultra-Plus slides. Deparaffinization and rehydration were performed in three changes of xylol and two changes of absolute ethanol followed by a graded series of ethanol descending from 90% to 50% and a final wash in triple distilled water for 5 min each. Pretreatment with amylase was performed by incubating a subgroup of samples in 0.1% amylase (Sigma-Aldrich, Munich, Germany) for 30 min at 37 °C, followed by two rinses for 5 min in triple-distilled water. All slides were immersed into 1% periodic acid for 5 min, followed by three rinses in triple-distilled water, and the incubation with the Schiff’s reagent (Sigma-Aldrich, Germany) for 15 min. Slides were washed three times for 1 min with sulfite-water (freshly prepared by first mixing 5 mL of 10% sodium disulfite (Merck, Darmstadt, Germany) with 5 mL of 1N hydrochloric acid and adding 100 mL tap water), followed by 15 min under tap water and 5 min in triple-distilled water. The sections were dehydrated in a graded series of ethanol (75–96%–2 × 100%) for 30 s each, followed by two times for 5 min in xylol, and mounted in non-aqueous medium.

### 4.4. Scoring of PAS Staining Intensity

Following the differential PAS staining, images of the entire tumor area were acquired under 100× magnification by light microscopy (Leica DMI 6000B, Wetzlar, Germany) without using a green filter. After background correction, the magenta color of the PAS reaction, which is a mixture of red and blue components, was intensified by subtracting the green component by adjusting the minimum green value to “104” in the Fiji software (Laboratory for Optical and Computational Instrumentation, University of Wisconsin-Madison, Madison, WI, USA, http://imagej.nih.gov/ij; version 1.52p). Image deconvolution was then performed using the Fiji software to separate the layers of PAS reaction and pigmentation with minimal overlap using the following, user-defined red (R), green (G), and blue (B) values: magenta for PAS (R1: 0.182, G1: 0.969, B1: 0.169); brown for the pigmentation (R3: 0.446, G3: 0.616, B3: 0.649); and blue-green for background (R2: 0.776, G2: 0.501, B2: 0.382). The gray value of the PAS layer was inverted to acquire higher pixel intensities against a dark background. The tumor area was marked on the original image and the integrated density (area × intensity) of the selected region was calculated by redirecting the measurement to the inverted PAS image. To determine the mean PAS intensity, the sum of the integrated densities was divided by the total area. The mean intensity of the amylase-treated sections was subtracted from the mean intensity of the corresponding, untreated tumor sample. The difference was expressed as the percentage of the intensity in the untreated tumor and taken as the estimated amount of amylase-sensitive glycogen. The total number of images quantified for the untreated and amylase-digested samples were *n* = 1428 and *n* = 1380, respectively.

### 4.5. Immunohistochemistry

Paraffin sections were rehydrated as described above. For antigen retrieval, sections were incubated in pre-heated 10 mM sodium citrate buffer, pH 6.0 with 0.025% Tween 20 for 20 min in a steam-cooker. After cooling to room temperature (RT) for 30 min, sections were rinsed three times for 10 min in phosphate-buffered saline (PBS) and incubated with sterile-filtered blocking buffer (3% BSA in 10 mM Tris-HCl, pH 7.5, 120 mM KCl, 20 mM NaCl, 5 mM EDTA, 0.1% Triton X-100) supplemented with 5% goat serum for 30 min at RT followed by the polyclonal rabbit primary antibodies against BAP1 (Abcam, Cambridge, UK; ab199396; 1:10 dilution in blocking buffer) overnight at 4 °C. The negative controls were incubated with the blocking buffer alone. Sections were then rinsed three times for 5 min with PBS, blocked in freshly prepared 3% hydrogen peroxide solution in PBS for 15 min, rinsed twice with PBS, and incubated with HRP-conjugated goat anti rabbit secondary antibodies (Jackson Immunoresearch; Cambridgeshire, UK; 111-035-003; 1:250 in blocking buffer) for 1 h at RT. Following three rinses for 5 min with PBS, sections were incubated for 10 min with the HRP green substrate that was freshly prepared by mixing 90 µL of HRP-Green chromogen with 1 mL of HRP-Green buffer as instructed by the manufacturer (42 Life Sciences, Bremerhaven, Germany). After rinsing for 5 min in triple-distilled water, nuclei were counterstained with nuclear fast red for 10 min. The slides were then briefly washed in triple-distilled water for 1 min, followed by the dehydration and mounting in non-aqueous medium as described above.

Images of the entire tumor area were acquired under 200× magnification using a light microscope (Leica). Quantification of the cytoplasmic and nuclear BAP1 expression was performed as described [89].

### 4.6. Immunomagnetic Enrichment of CMCs

CMCs were isolated within three hours of venous blood collection as described [90]. Isolated cells were processed as cytospins on to Superfrost Plus slides and stored at −20 °C until use.

### 4.7. Immuno-FISH

Processing of frozen cytospins and paraffin sections for Immuno-FISH was performed as described [8], using the primary antibodies against Melan-A (Abcam, ab210546; 1:100 in blocking buffer) and Alexa 488-conjugated goat anti-rabbit secondary antibodies (Abcam, ab150077; 1:100 in blocking buffer) for the latter group. Cells were analyzed by fluorescence microscopy (Leica DMI 6000B) using the appropriate filter sets (A4: Ex: 360/40, Em: 470/40 nm; L5: Ex: 460/40, Em: 527/30 nm; Cy3: Ex: 545/30, Em: 610/75 nm). Images were acquired with a monochrome digital camera (DFC 350 FX; Leica) attached to the microscope and the Leica Application Software (Advanced Fluorescence 2.3.0, build 5131).

The copy number of chromosome 3 in the tumor samples was determined in the cells positive for Melan-A. Signal quantification was performed by calculating both the percentage of monosomy-3-positive cells and the chromosomal index as described [91]. For the former method, the percentage of cells with monosomy-3 was calculated in a minimum of *n* = 203 non-overlapping nuclei within a given area. The chromosomal index was determined by counting the total signals for chromosome 3 and dividing by the number of nuclei in a given area. The median results were calculated for both quantifications (Median percentage of monosomy-3 cells: 31.94%, range: 11.65–85.46%; median chromosomal index: 1.01, range: 0.65–1.63). Tumors with a percentage of monosomy-3 cells that was equal to or above the median or a chromosomal index equal to or below the median obtained the score 1. The tumors that received the score 1 for both parameters were classified as having “Monosomy-3.” Samples that received the score 0 with both methods were classified as “Disomy-3,” and the tumors that received the score 1 with only one method were classified as intermediate. For the subsequent statistical analysis, the disomy-3 and intermediate tumors were combined and collectively defined as “Disomy-3 tumors.”

### 4.8. Statistical Analysis

Data sets were analyzed using the NCSS statistical software (Version 19.0.3, NCSS, LLC, Kaysville, UT, USA) under Windows 10. Mann–Whitney U test was performed for evaluating the association of numerical parameters with the binary variables. Kruskal–Wallis test was applied for the variables that can be divided into three or more subgroups. Pearson’s Chi Square test was performed to analyze the proportion of categorical variables. P values less than 0.05 were considered as significant. Bonferroni correction was performed by multiplying the P values with the total number of comparisons in the multiple testing of gene expression.

## 5. Conclusions

In conclusion, our findings provide the first insight into the monosomy-3-dependent alterations in the glycogen metabolism of UM cells, which may be providing a growth advantage particularly in the male patients under hyperglycemic conditions. Prevention of insulin resistance and excessive glucose release from the liver might therefore be an attainable and immediately available therapeutic approach to maintain the dormancy of the UM cells that were released into the circulation or have formed micrometastases in the liver, which urgently deserves further attention.

## Figures and Tables

**Figure 1 cancers-12-02101-f001:**
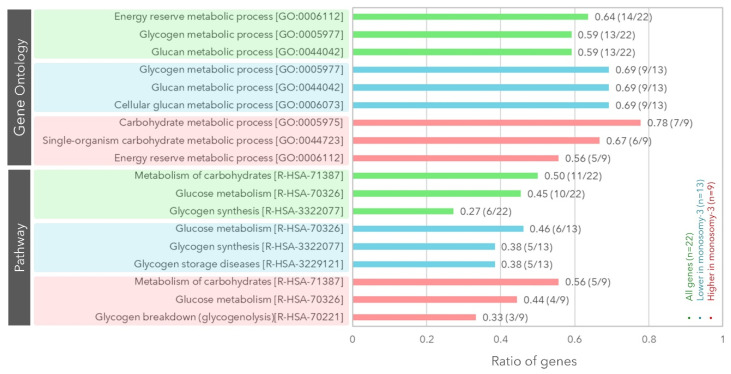
Gene set enrichment analysis demonstrating the gene ontology terms and biological pathways that were over-represented among the differentially expressed genes in the monosomy-3 tumors. The differentially expressed genes (*n* = 22) are collectively presented in green, whereas the down- and up-regulated genes in the monosomy-3 tumors are indicated in blue vs. red, respectively. The *y*-axis lists the Gene Ontology terms and Reactome pathways that were most enriched among the subgroups of genes, while the *x*-axis represents the ratio of genes within a subgroup that were mapping to these functional classifications. The ratio and number of genes are stated adjacent to the bar plots. The accession numbers for the Gene Ontology terms and Reactome pathways are indicated in square brackets. All the *p*-values and false discovery rates were <0.001.

**Figure 2 cancers-12-02101-f002:**
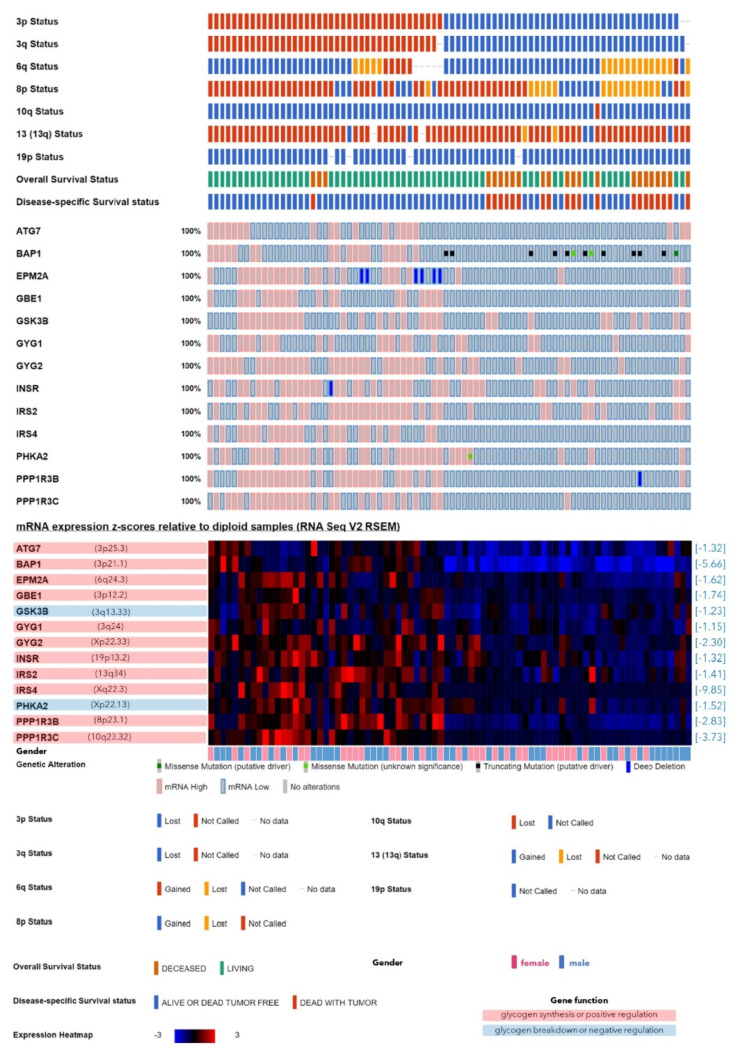
Genes involved in glycogen metabolism that exhibited a lower expression in the monosomy-3 tumors of the TCGA-UM cohort (*n* = 80 patients) regardless of the gene copy number. The tumor samples are represented by the columns, which were aligned according to the copy numbers of chromosome 3p and 3q (red: normal, blue: loss) in the uppermost two rows. Tumors with the loss of both 3p and 3q were defined as having monosomy-3, while this data was missing for *n* = 3 samples. The copy number of the chromosomes that harbor the remaining genes of interest, as well as the survival status are also presented in the upper rows, whereas the middle rows demonstrate the presence of genetic alterations. The expression heatmap in the lower rows was generated from the mRNA z-scores, with blue and red indicating mRNA levels that were up to three standard deviations lower or higher than the mean, respectively, whereas black denotes an expression at the mean. The loci of the analyzed genes are stated in parentheses next to the gene symbol in the heatmap. The fold changes are presented in square brackets on the right-hand side of the heatmap. Genes that function as positive versus negative regulators of glycogen synthesis are highlighted with a red or blue background, respectively. Patient gender is indicated underneath the heatmap.

**Figure 3 cancers-12-02101-f003:**
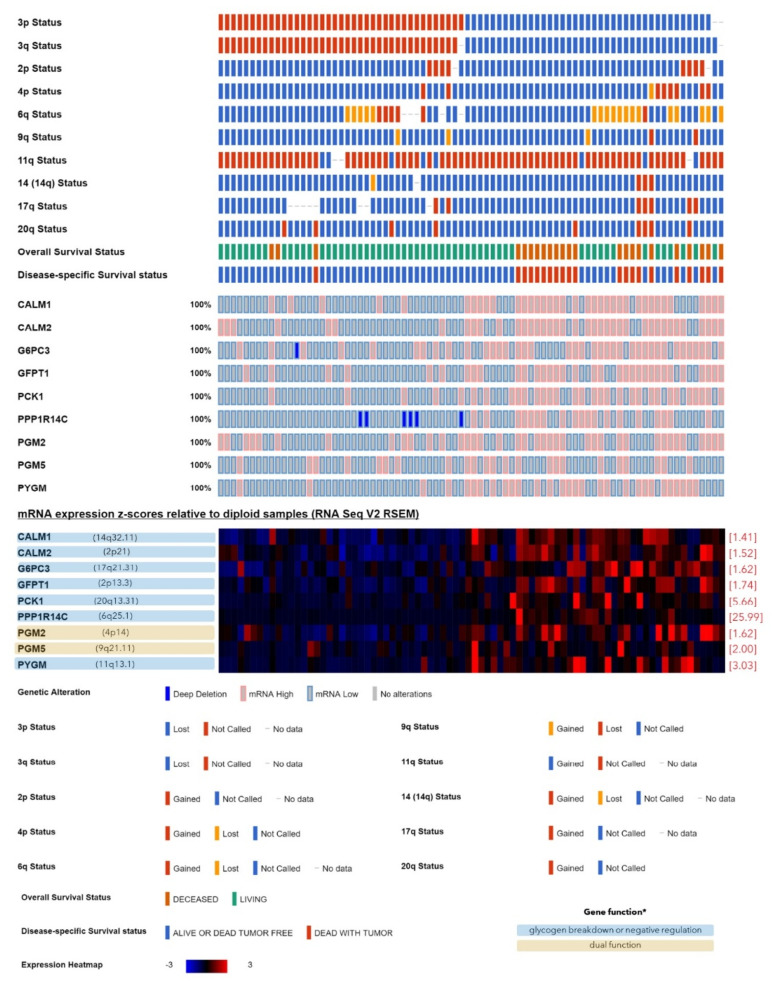
Genes involved in glycogen metabolism that exhibited a higher expression in the monosomy-3 tumors of the TCGA-UM cohort (*n* = 80 patients) regardless of the gene copy number. The tumor samples, which are represented by the columns, were aligned according to the copy numbers of chromosome 3p and 3q (red: normal, blue: loss) in the uppermost two rows. Tumors with the loss of both 3p and 3q were defined as having monosomy-3, while this data was incomplete for *n* = 3 samples. The copy number of the chromosomes that harbor the remaining genes of interest, as well as the survival status are also illustrated in the upper rows, whereas the middle rows demonstrate the presence of genetic alterations. The expression heatmap in the lower rows was established from the mRNA z-scores, with blue and red indicating mRNA levels that were up to three standard deviations lower or higher than the mean, respectively, whereas black denotes an expression at the mean. The loci of the analyzed genes are stated in parentheses next to the gene symbol in the heatmap. The fold changes are listed in square brackets on the right-hand side of the heatmap. Genes that negatively regulate the glycogen production are highlighted with a blue background whereas the genes that are involved in both the synthesis and breakdown of glycogen are indicated in yellow.

**Figure 4 cancers-12-02101-f004:**
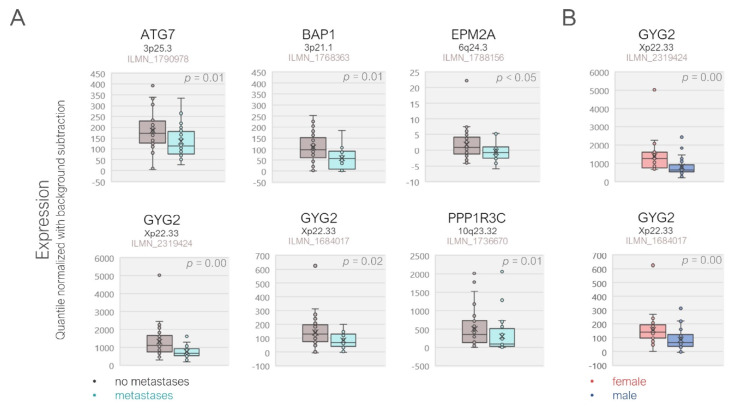
Differential gene expression in the validation cohort demonstrated by the scatter box plots. (**A**) The patients with metastases exhibited a significantly lower expression of the ATG7, BAP1, EPM2A, GYG2, and PPP1R3C genes. (**B**) GYG2 transcripts were also significantly lower in the tumors of the male versus female patients. *p*-Values were analyzed by the Mann–Whitney U test. The gene loci and the probe identifiers of the microarray are indicated underneath the gene symbols.

**Figure 5 cancers-12-02101-f005:**
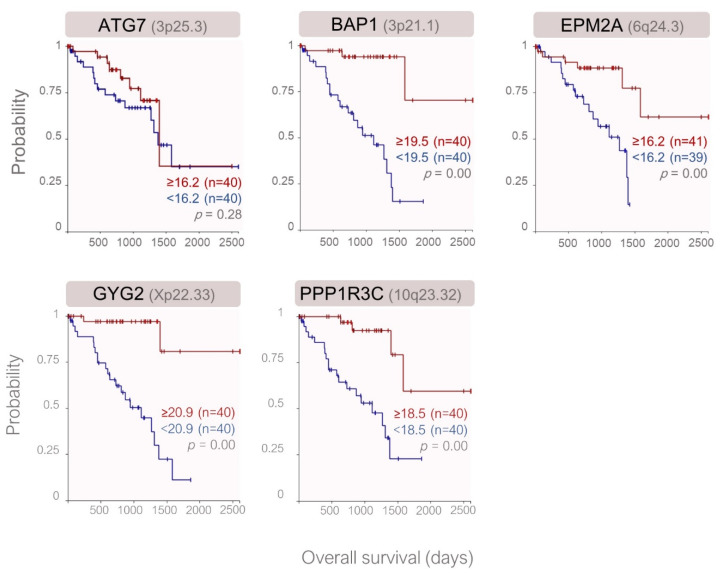
Kaplan–Meier curves demonstrating the probability of overall survival with respect to the expression of the validated genes in the primary UM samples of the TCGA study. The median mRNA level of each gene was taken as the cut-off value. Gene loci are indicated in parentheses following the gene symbol. All the presented genes function as positive regulators of glycogen synthesis. *p*-Values were determined by the log-rank test.

**Figure 6 cancers-12-02101-f006:**
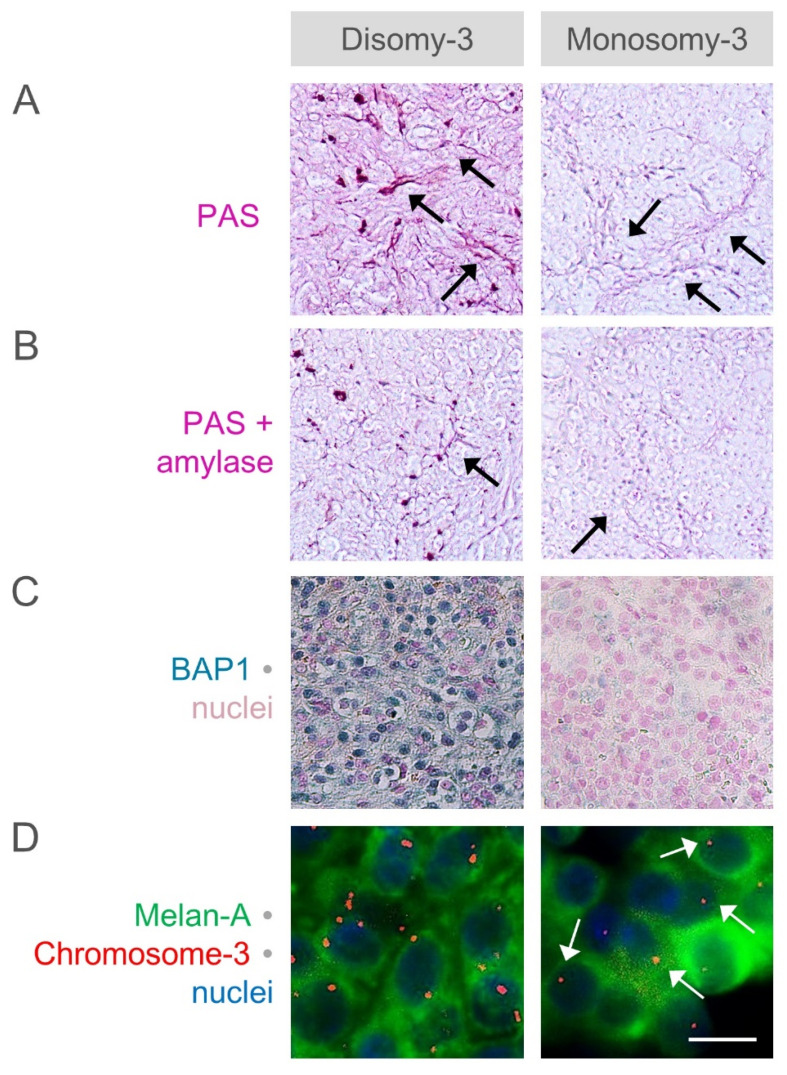
Estimated levels of amylase-sensitive glycogen with regard to the monosomy-3 status and nuclear BAP1 expression. The difference in the intensity of the periodic acid-Schiff (PAS) stainings (**A**) without and (**B**) with amylase pretreatment was taken as the amount of amylase-sensitive glycogen. The amylase pretreatment has also led to a weakening in the staining of the extracellular arcs (black arrows). Original magnification: 100×. (**C**) Expression of BAP1 was analyzed by immunohistochemistry and staining of the nuclei with nuclear fast red. Original magnification: 200×. (**D**) The copy number of chromosome-3 (red) together with the melanoma marker protein Melan-A (green) was determined by performing a combined immunostaining with fluorescent in situ hybridization (Immuno-FISH) on the paraffin sections. Nuclei were counterstained in blue with 4′,6-diamidino-2-phenylindole. Arrows indicate several cells with monosomy-3. Scale bar = 10 µm.

**Figure 7 cancers-12-02101-f007:**
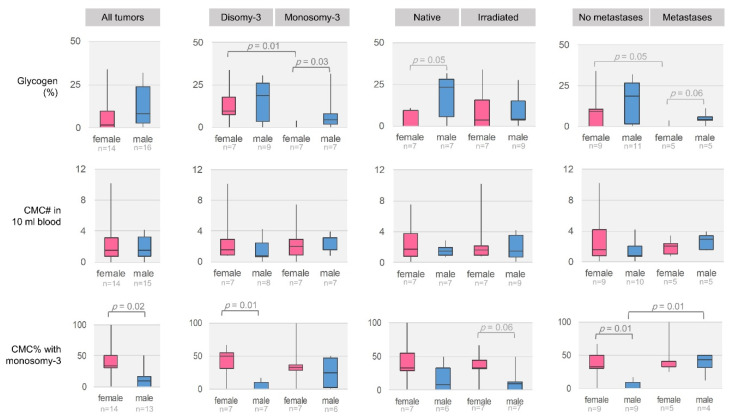
Comparison of the glycogen levels and CMCs in the female versus male patients with regard to the monosomy-3 status, irradiation, and metastases. *p*-Values were determined by the Mann-Whitney U test. N = number.

**Table 1 cancers-12-02101-t001:** Human genes involved in glycogen synthesis or breakdown that are differentially expressed in the monosomy-3 tumors of the uveal melanoma (UM) cohort of the Cancer Genome Atlas (TCGA) study.

Symbol	Name	Locus	Function in Glycogen Synthesis [References]	Expression in Monosomy-3 Tumors			
				All Patients (*n* = 77)		Omission of the Patients with an Aberrant Gene Copy Number	
				Fold Change	*p* ^†^	Fold Change	*p* ^†^
GBE1	1,4-alpha-glucan branching enzyme 1	3p12.2	de novo synthesis [15,16]	−1.74	0.00	n.a.	n.a.
GYG1	glycogenin 1	3q24	de novo synthesis [15,16]	−1.15	0.01	n.a.	n.a.
GYG2	glycogenin 2	Xp22.33	de novo synthesis [15,16]	−2.30	0.00	−2.46	0.00
ATG7	autophagy related 7	3p25.3	positive [15]	−1.32	0.00	n.a.	n.a.
BAP1	BRCA1 associated protein 1	3p21.1	positive [22]	−5.66	0.00	n.a.	n.a.
EPM2A	EPM2A glucan phosphatase, laforin	6q24.3	positive [16]	−1.62	0.00	−1.74	0.00
INSR	insulin receptor	19p13.2	positive [31]	−1.32	0.00	−1.32	0.00
IRS2	insulin receptor substrate 2	13q34	positive [32]	−1.41	0.00	−1.32	0.00
IRS4	insulin receptor substrate 4	Xq22.3	positive [33]	−9.85	0.00	−8.57	0.02
PPP1R3B	protein phosphatase 1 regulatory subunit 3B	8p23.1	positive [15,16]	−2.83	0.00	−2.30	0.00
PPP1R3C	protein phosphatase 1 regulatory subunit 3C	10q23.32	positive [15,16]	−3.73	0.00	−3.73	0.00
CALM1	calmodulin 1	14q32.11	negative [16]	1.41	0.00	1.41	0.00
CALM2	calmodulin 2	2p21	negative [16]	1.52	0.00	1.52	0.00
G6PC3	glucose-6-phosphatase catalytic subunit 3	17q21.31	negative [16]	1.62	0.00	1.74	0.02
GFPT1	glutamine--fructose-6-phosphate transaminase 1	2p13.3	negative [34]	1.74	0.00	1.74	0.00
GSK3B	glycogen synthase kinase 3 beta	3q13.33	negative [15,23]	−1.23	0.00	n.a.	n.a.
PCK1	phosphoenolpyruvate carboxykinase 1	20q13.31	negative [35]	5.66	0.00	6.96	0.00
PHKA2	phosphorylase kinase regulatory subunit alpha 2	Xp22.13	negative [16]	−1.52	0.00	−1.52	0.03
PPP1R14C	protein phosphatase 1 regulatory inhibitor subunit 14C	6q25.1	negative [36]	25.99	0.00	27.86	0.00
PYGM	glycogen phosphorylase, muscle associated	11q13.1	negative [15,16]	3.03	0.00	3.25	0.00
PGM2	phosphoglucomutase 2	4p14	dual [16,38]	1.62	0.00	1.52	0.02
PGM5	phosphoglucomutase 5	9q21.11	dual [16,38]	2.00	0.02	2.14	0.00

n.a.: not analyzed; TCGA: The Cancer Genome Atlas; UM: Uveal melanoma. ^†^ Mann–Whitney U test and Bonferroni adjustment. Fold changes were calculated using the median gene expression. The negative fold change values indicate the downregulation of genes in the monosomy-3 tumors. The values for gene expression and the sample sizes are presented in detail in Appendix A.

**Table 2 cancers-12-02101-t002:** Association of the glycogen levels with the histopathological and clinical factors (follow-up time: 2–9 years).

	All Patients *n* = 30	High Glycogen (≥Median) *n* = 15	Low Glycogen (<Median) *n* = 15	*p* *
Age				
Median (range)	69 (39–83)	73 (50–83)	63 (39–81)	0.07
Gender (*n*, %)				
Female	14 (46.7)	6 (40.0)	8 (53.3)	0.46
Male	16 (53.3)	9 (60.0)	7 (46.7)	
Eye (*n*, %)				
Right	13 (43.3)	5 (33.3)	8 (53.3)	0.27
Left	17 (56.7)	10 (66.7)	7 (46.7)	
Irradiation (*n*, %)				
No	15 (50.0)	8 (53.3)	7 (46.7)	0.72
Yes	15 (50.0)	7 (46.7)	8 (53.3)	
Tumor size in mm (Median, range)				
Basis 1	12.9 (1.4–18.6)	12.8 (8.5–18.6)	13.4 (1.4–17.5)	0.90
Basis 2	12.8 (1.4–21.8)	14.0 (6.2–21.8)	12.0 (1.4–19.0)	0.29
Prominence	7.9 (0.5–17.8)	9.0 (0.5–17.8)	6.6 (0.9–12.8)	0.17
Monosomy-3 in primary tumor (*n*, %)				
No	16 (53.3)	12 (80.0)	4 (26.7)	**0.00**
Yes	14 (46.7)	3 (20.0)	11 (73.3)	
BAP1 expression, cytoplasmic (*n*, %)				
High	18 (60.0)	11 (73.3)	7 (46.7)	0.14
Low	12 (40.0)	4 (26.7)	8 (53.3)	
BAP1 expression, nuclear (*n*, %)				
High	11 (36.7)	9 (60.0)	2 (13.3)	**0.01**
Low	19 (63.3)	6 (40.0)	13 (86.7)	
CMC presence (*n*, %) ^†^				
No	2 (6.9)	1 (7.1)	1 (6.7)	0.96
Yes	27 (93.1)	13 (92.9)	14 (93.3)	
CMC number/10 mL blood				
Median (range)	1.6 (0.0–10.2)	1.6 (0.0–10.2)	1.5 (0.0–7.5)	0.59
Monosomy-3 in CMCs (*n*, %) ^††^				
No	8 (29.6)	5 (35.7)	3 (23.1)	0.47
Yes	19 (70.4)	9 (64.3)	10 (76.9)	
% CMCs with monosomy-3				
Median (range)	30.4 (0.0–100)	23.6 (0.0–66.7)	31.8 (0.0–100)	0.75
Metastases (*n*, %)				
No	20 (66.7)	13 (86.7)	7 (46.7)	**0.02**
Yes	10 (33.3)	2 (13.3)	8 (53.3)	

%: percentage; *n* = number, CMC= circulating melanoma cell. * Mann–Whitney U test for the numerical and Pearson’s Chi-Square test for the categorical parameters. ^†^ Data missing from one patient. ^††^ Data missing from three patients.

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
