# Peer review of "Metastasis of Uveal Melanoma with Monosomy-3 Is Associated with a Less Glycogenetic Gene Expression Profile and the Dysregulation of Glycogen Storage"

_cancers, 2020, doi:10.3390/cancers12082101_

Round 1
Reviewer 1 Report
The authors have addressed my all concerns.
Author Response
Reviewer 1:
The authors have addressed my all concerns.
Author's Response:
Thank you very much for kindly approving our previous response and your valuable suggestions that have helped us to improve our manuscript.
Best regards
Reviewer 2 Report
Comment:
Metastasis of uveal melanoma is associated with a less glycogenetic gene expression profile and the dysregulation of glycogen storage is an interesting paper where authors have delineated the biochemical reason behind the worst prognosis. So, it is suitable for publication. However, authors used in their studies Uveal melanoma with monosomy-3 condition. So this should be known to the readers. So they have to make following minor corrections:
- Title should be modified a bit to reflect this correlation of uveal melanoma with monosomy-3. So, it should be rewritten as Metastasis of uveal melanoma with monosomy-3 is associated with a less glycogenetic gene expression profile and the dysregulation of glycogen
- In the introduction part authors should highlight this correlation of uveal melanoma with monosomy-3 and worst prognosis
Author Response
Reviewer's Comments:
Metastasis of uveal melanoma is associated with a less glycogenetic gene expression profile and the dysregulation of glycogen storage is an interesting paper where authors have delineated the biochemical reason behind the worst prognosis. So, it is suitable for publication. However, authors used in their studies Uveal melanoma with monosomy-3 condition. So this should be known to the readers. So they have to make following minor corrections:
- Title should be modified a bit to reflect this correlation of uveal melanoma with monosomy-3. So, it should be rewritten as Metastasis of uveal melanoma with monosomy-3 is associated with a less glycogenetic gene expression profile and the dysregulation of glycogen
- In the introduction part authors should highlight this correlation of uveal melanoma with monosomy-3 and worst prognosis
Author's Response
-
Thank you very much for your kind remarks and recommending our study for publication. As suggested, we have performed the following minor corrections:
- The title was rewritten as "Metastasis of uveal melanoma with monosomy-3 is associated with a less glycogenetic gene expression profile and the dysregulation of glycogen storage".
- We have tried to underline the correlation of monosomy-3 with the worst prognosis in UM patients by adding the highlighted sentence into the Introduction (Pages 1 and 2, Lines 40-44):
“The most critical prognostic factor for UM is the loss of one copy of chromosome 3 (monosomy-3), with the occurrence of metastatic disease almost exclusively in the patients having this anomaly in their primary tumor [4,6,7]. Likewise, the presence of monosomy-3 in the circulating melanoma cells (CMC) and metastasized UMs was associated with a higher metastatic risk and disease progression rate, respectively [8,9]. Monosomy-3 therefore serves as an independent prognostic factor for shorter survival following the diagnosis of metastases and the UMs with this anomaly are classified as high-risk tumors [4,7]. “
This manuscript is a resubmission of an earlier submission. The following is a list of the peer review reports and author responses from that submission.
Round 1
Reviewer 1 Report
First of all, mortality or incidence of cutaneous melanoma is higher in males than in females. In fact, mortality rate in males are generally higher in males than in females in all types of cancers. There are several reasons for that. Warberg effect is also observed in other solid tumors not only with UM. Authors can help me to decide, if they can answer following questions
- Whether monosomy-3 is a cause for UM or a consequence of UM?
- Have they encountered UM without monosomy-3? (because they have discussed about that in their text)
- Have authors encountered any case where there is UM and type 2 diabetes (because in T2D, there is hyperglycemia and also insulin resistance)? If so, did they find same result in that case?
- In non UM, monosomy-3 case, whether they observed same glycogenolytic events?
- Do authors have information about PAS staining and glycogenolytic events in other solid tumors, other than melanoma?
Reviewer 2 Report
In this manuscript, the authors analyzed the expression of 67 genes involved in glycogen metabolism from UM TCGA data and then validated 22 differentially expressed genes between monosomy 3 and disomy 3 tumors in another cohort from the public database. Furthermore, they examined the glycogen levels in the primary UM of their patients with or without monosomy 3 status as well as association of the glycogen levels with the histopathological and clinical factors. They concluded that UMs with monosomy-3 displayed a reduction of glycogen levels and more insulin-resistant gene expression profile associated with the metastases. Additionally, they claimed that dysregulation of glycogen metabolism may be different between male and female UM patients with monosomy-3. Monosomy-3 has been shown to be associated with UM metastasis. The underlying mechanism is still unclear. If the findings from this study are correct, it has significant clinical implication. However, the main weakness in this paper is that the data are a little biased and not strong enough to support their conclusions and all figures are not clearly presented. I have the following comments:
- The data about 22 differentially expressed genes identified from UM TCGA samples is not convincing. The fold changes for most of genes between disomy 3 and monosomy tumors are very small. For example, for gene GYG1, mean expression value is 18.4 for disomy 3 compared to 18.2 for monosomy. Thus, it is not sure whether there is any significant biological meaning. Fold changes are very important data and should be presented in main figures. The authors should also perform non-biased Gene Set Enrichment Analysis (GSEA) to validate their findings.
- Most of figures and tables are too complicated and very hard to understand. It will be helpful if they use scatter plot to present their data.
- Some important data should be presented in details. For example, the authors claimed there are significant difference in glycogen levels between male and female monosomy-3 tumors. The author should show all PAS staining pictures for those tumors either in main figures or supplemental figures.
- The analysis is a little biased, thus the conclusion is not very convincing. For example, for figure 3, the authors only showed Kaplan-Meier curves for only 4 genes to demonstrate that the reduced probability of overall survival with respect to the lower expression of the validated genes in the primary UM samples of the TCGA study.